# Vaginal, Cervical and Uterine pH in Women with Normal and Abnormal Vaginal Microbiota

**DOI:** 10.3390/pathogens10020090

**Published:** 2021-01-20

**Authors:** Malene Risager Lykke, Naja Becher, Thor Haahr, Ebbe Boedtkjer, Jørgen Skov Jensen, Niels Uldbjerg

**Affiliations:** 1Department of Obstetrics and Gynecology, Aarhus University Hospital, DK-8200 Aarhus, Denmark; THOHAA@rm.dk (T.H.); uldbjerg@clin.au.dk (N.U.); 2Department of Molecular Biology and Genetics, Aarhus University Hospital, DK-8200 Aarhus, Denmark; nbecher@dadlnet.dk; 3The Fertility Clinic, Skive Regional Hospital Denmark, DK-7800 Skive, Denmark; 4Department of Biomedicine, Aarhus University, DK-8000 Aarhus, Denmark; eb@biomed.au.dk; 5Research Unit for Reproductive Microbiology, Statens Serum Institut, DK-2300 Copenhagen, Denmark; JSJ@ssi.dk

**Keywords:** abnormal vaginal microbiota, bacterial vaginosis, cervical mucus plug, female reproductive tract, vaginal pH, vaginal physiology, uterine pH

## Abstract

Introduction: Healthy women of reproductive age have a vaginal pH around 4.5, whereas little is known about pH in the upper genital tract. A shift in the vaginal microbiota may result in an elevated pH in the upper genital tract. This might contribute to decreased fertility and increased risk of preterm birth. Therefore, we aimed to measure pH in different compartments of the female genital tract in both nonpregnant and pregnant women, stratifying into a normal and abnormal vaginal microbiota. Material and methods: In this descriptive study, we included 6 nonpregnant, 12 early-pregnant, and 8 term-pregnant women. A pH gradient was recorded with a flexible pH probe. An abnormal vaginal microbiota was diagnosed by a quantitative polymerase chain reaction technique for *Atopobium vaginae*; *Sneathia sanguinegens*; *Leptotrichia amnionii*; bacterial vaginosis-associated bacterium 1, 2, 3, and TM7; and *Prevotella* spp. among others. Results: In all participants we found the pH gradient in the lower reproductive canal to be most acidic in the lower vagina and most alkaline in the upper uterine cavity. Women with an abnormal vaginal microbiota had an increased pH in the lower vagina compared to the other groups. Conclusions: There is a pronounced pH gradient within the female genital tract. This gradient is not disrupted in women with an abnormal vaginal microbiota.

## 1. Introduction

Disturbance in the normal vaginal flora affects the health of a woman and, in pregnancy, her fetus and newborn child. A switch from what is considered a healthy vaginal microbiome dominated by *Lactobacillus* spp. to facultative and strict anaerobes causes bacterial vaginosis (BV) [1]. BV is the predominant cause of genital complaints in women of childbearing age worldwide [2]. It is well established that the pH within the lower vagina is around 4.5 in healthy women of fertile age [1,2]. However, we know little about pH in the upper compartments of the female genital tract, including the cervical canal and the uterine cavity, neither in healthy women nor in women with bacterial vaginosis (BV).

One of many reasons to describe the pH in the female genital tract is the possible association of this variable to BV, a dysbiosis of the vaginal microbiota presenting with a loss of lactic acid producing *Lactobacillus* spp. and hence an increase in pH. This change in pH is easy to detect with simple tools/measures. BV is, among others, associated with pelvic inflammatory disease [3], infertility [4], spontaneous abortion [5], and preterm birth [6]. It has been established that the composition of the vaginal microbiota during pregnancy changes as function of gestational age with an increase in *Lactobacillus* spp., and the stability of the microbiota becomes more stable during pregnancy [7,8]. Likely, pH-associated pathways include (1) disrupted local immunological properties [9,10,11] caused by a pH dependence of the antimicrobial components in the cervical mucus plug [9,10], (2) a changed behavior of the mucins that determine the visco-elastic properties of the cervical mucus [9,11], and (3) the formation of a biofilm within the uterine cavity [12,13].

BV is traditionally diagnosed by Gram staining of vaginal secretion using a validated morphological grading system, e.g., the Nugent score [14]. However, the Nugent score classifies women into three categories with an intermediate group that is difficult to manage clinically and often classified with normal vaginal microbiota. In contrast, next generation sequencing methods have enabled the classification of the vaginal microbiome according to so-called community state types (CSTs) [15]. Four CSTs are dominated by *Lactobacillus* spp. whereas the diversity type (CST IV) comprises most women with BV [2,15]. CST IV is mainly dominated by *Gardnerella* spp. and *Atopobium vaginae*, and detection of these species by quantitative polymerase chain reaction (qPCR) above a certain threshold has been shown to accurately detect BV and to classify the intermediate group into a normal vaginal microbiota (NVM) or abnormal vaginal microbiota (AVM) [16].

We therefore categorized nonpregnant, early-pregnant, and at-term-pregnant women into NVM and AVM participants and measured their pH in the lower and the upper vagina, in the lower and the upper part of the cervical canal, and also in the nonpregnant women in the lower and the upper part of the uterine cavity.

## 2. Results

We identified AVM in three of six nonpregnant women, in five of 12 early-pregnant women and in none of eight at-term-pregnant women. There was compliance between qPCR and Nugent score. For further details concerning qPCR results, Nugent classification, and detailed pH values please see the Appendix A.

The effect of AVM on pH was significant only in the lower vagina, much less marked and not statistically significant in the upper vagina, and not measurable at all in the cervical canal or the uterine cavity. Among the nonpregnant women with NVM, we demonstrated a striking pH gradient with a median value of 3.9 (range: pH 3.6–4.3) in the lower vagina, 5.7 (5.2–6.3) in the upper vagina, a small but significant gradient within the cervical canal, and not less than 7.7 (7.5–7.8) in the upper uterine cavity (Figure 1b and Figure 2A). In early pregnancy and at-term pregnancy, the values in the vagina were rather close to those from nonpregnant women; however, for at-term pregnancy, the values within the cervical canal were decreased by about 1.0 pH unit (Figure 2A).

In nonpregnant women, AVM was associated with an increased pH in the lower vagina (median pH 4.7). However, this was not the case in the upper vagina, the cervical canal, and the uterine cavity (Figure 2B). The same pattern was seen among early-pregnant AVM participants (Figure 2C).

Five of the twelve early-pregnant participants had misoprostol (Cytotec, 0.4 mg) two hours prior to the pH assessment. Their median pH in the lower vagina was 4.1 (range: 3.3 to 4.6) vs. 4.0 (range: 3.6 to 4.6) in the untreated participants. In the cervical canal, however, misoprostol increased the pH to 7.2 (range: 6.8 to 7.6) vs. 6.3 (range: 4.4 to 7.4) (*p* < 0.01). Two of the three AVM-participants had misoprostol.

## 3. Discussion

This study demonstrated a pronounced pH gradient in the female genital tract. In nonpregnant women with NVM, the median pH was 3.9 in the lower vagina, 5.7 in the upper vagina, and not less than 7.7 in the upper uterine cavity. Furthermore, it was remarkable that the pH in the lower part of the cervical canal decreased from 6.3 in early pregnancy to 5.4 at term, nearly a 10-fold increase in the H+ concentration. Finally, the effect of AVM on the pH was significant only in the lower vagina, much less marked and not statistically significant in the upper vagina, and not measurable at all in the cervical canal or the uterine cavity.

It is a strength of our study that we conducted the pH measurements continuously within a few minutes. Despite that, we cannot exclude that changes in the partial pressure of CO_2_ in the vagina during a gynecological examination affected the pH in the vagina [16]. If so, the loss of CO_2_ would probably be most prominent from the vagina and increase the pH here, i.e., resulting in an underestimation of the pH gradient along the reproductive tract. It is also a strength of our study that we used a modern qPCR-based technique for AVM diagnosis [16,17,18,19]; however, future studies should consider not only qPCR-based determination of bacterial load but also 16S rRNA gene or whole genome microbiome studies to determine composition at various levels of the vagina together with the pH measurements.

The size of the study is a limitation as it did not allow us to stratify the nonpregnant participants according to their menstrual cycle or their use of hormonal contraceptives. Further studies involving a larger number of subjects are necessary to draw more firm conclusions. Even though the microbiome seems to vary within the vagina [20,21,22,23], it was not an aim of our study to determine the bacterial composition of the vagina. Concerning the effect of misoprostol on pH, it is a weakness that we have not measured the pH gradient in the time span from the consumption to the time of pH measurement, i.e., we were not able to compare the time-response with that known for acid production in the ventricle. Our cohort consisted mainly of Caucasians. Therefore, care should be taken generalizing to other ethnic groups as microbial communities vary among ethnic groups [2]. It is also described that these communities, community state groups (CTS), depict higher pH in the CTS IV group (equivalent to AVM) than in groups I, II, and III (mainly NVM) [2].

In the vagina, the pronounced compartmentalization in terms of pH may depend not only on the vaginal microbiota but also on acid–base transporters, such as Na^+^, HCO_3_^−^-cotransporters, Na^+^/H^+^-exchangers, and epithelial proton pumps, which are abundant throughout the vaginal epithelium [24]. During pregnancy, the levels of estrogen exhibit at least two relevant effects: they increase the glycogen levels in the vagina, which is substrate for the *Lactobacillus* spp.; and they affect the acid–base transport across the epithelium [24,25]. This may contribute to the findings in previous articles showing that the pregnant vaginal microbiome is equally rich but less diverse than the nonpregnant microbiome [2,8]. Our study shows the same pattern expressed in terms of pH: the term-pregnant pH gradient is less pronounced than the nonpregnant gradient.

In the cervical canal, the pH gradient may be associated to an equivalent *Lactobacillus* spp. gradient [11]. However, the acid–base transporters may be even more important as the bacterial load within the cervical canal is much lower than that of the vagina. The pregnancy-associated decrease of pH in the cervix (Figure 2A) may stabilize the cervical mucus plug [26,27] and affect the antimicrobial properties of the plug [9,10,27,28].

In the uterine cavity of the nonpregnant women, the pH was close to neutral and in the upper part of the cavity even alkaline (range: 7.0–7.8). In this compartment, acid–base transport across the epithelium plays a role in cows and mice [29,30], whereas the biomass including *Lactobacillus* [12] is probably too small to affect the pH, even though it may influence fertility [4,12].

Our results raise several clinically relevant questions and considerations. First, the uterine pH is not affected by the increased vaginal pH in women with AVM or BV. Secondly, when screening for BV using a pH glove [31,32,33] or paper strips [34], it seems important to specify whether measurements are obtained from the lower or the upper vagina. Thirdly, one can hypothesize that the moving direction of the spermatozoa in their journey from the upper vagina to the uterus towards the oocyte is controlled by the pH gradient. This suggestion is supported by the finding that bovine sperm transport is influenced by uterine pH [30]. To verify this hypothesis it would require evidence of pH changes before and after ovulation and a uterine contractility gradient to support it further. A larger study involving a larger group of women in different phase of menstrual cycle is necessary to test this hypothesis.

## 4. Materials and Methods

### 4.1. Study Participants

We enrolled 26 women (25 Caucasian, 1 Indian), consisting of six nonpregnant women having a laparoscopic sterilization (mean age 39.2 years), twelve early-pregnant women having termination of pregnancy (gestational age range: 6 + 5–10 + 5 weeks, mean age 35.5 years), and eight term-pregnant women having induction of labor (37 + 0–41 + 5 weeks, mean age 34.0 years). Laparoscopic sterilization and termination of pregnancy was conducted at Randers Regional Hospital, Randers, Denmark, whereas inducted labor was performed at Aarhus University Hospital, Aarhus. The termination of pregnancy was conducted according to the Danish law of “free abortion” until gestational age 12 weeks, i.e., they were not indicated by abnormal findings. The exclusion criteria were symptoms of BV, vaginal or menstrual bleeding, rupture of membranes, reported or visual cervical mucus plug discharge, intercourse within 24 h of the procedure, and treatment with antibiotics less than one week prior to the procedure. We performed ultrasonic measurements during the first two laparoscopic sterilizations, but it was not always possible to have access to ultrasound during each pH measurement since this was not standard during the surgical procedure. The Regional Committee on Health Research Ethics (1-10-72-112-14) and the Danish Data Protection Agency (2014-112-14) approved the protocol, and the women were informed orally about the procedure and gave their written informed consent prior to the procedure.

### 4.2. Microbiology

We obtained a vaginal flocked swab sample from the posterior fornix in ESwab transport medium (Copan Italia, Brescia, Italy), which was kept at 5 °C until freezing at –80 °C no later than 3 h after sampling. One investigator (TH) who was blinded to the clinical data performed the BV diagnostics: “normal” (Nugent’s score 1–3), “intermediate” (Nugent’s score 4–6), and “BV” (Nugent’s score 7–10) [14]. AVM was diagnosed by real-time qPCR for *Atopobium vaginae* and 3 additional bacteria associated with BV above a threshold bacterial load as described by Datchu et al. [16]. Quantitative PCRs were performed on vaginal swabs for selected BV associated bacteria, including *A*. *vaginae*; *Sneathia sanguinegens; Leptotrichia amnionii;* bacterial vaginosis-associated bacterium 1, 2, 3 and TM7 (BVAB 1, 2, 3, and TM7); *Megasphaera* type 1 and 2; *Eggerthella*-like bacterium; *Mobiluncus curtisii* and *Mobiluncus mulieris*; *Lactobacillus iners*; *Gardnerella vaginalis*; *Mycoplasma hominis*; *Ureaplasma parvum*; *Ureaplasma urealyticum*; *Finegoldia magna*; and *Prevotella* spp. The *Prevotella* probe used was a genus-wide probe covering multiple *Prevotella* species, including *Prevotella bivia.* Primers and specific controls are described in detail in this paper referenced [16]. These four bacterial species were found to be the optimal bacteria for predicting BV. All samples were run in the same lab and with the same methods as described in the paper by Datchu et al. Briefly, the *A. vaginae* qPCR assay was run with a TaqMan probe, whereas the *Prevotella* spp., *Megasphaera*, and BVAB assays were run with SYBR Green assays. All assays were run on the 7500 ABI instrument. The reagents used and the cycling conditions have been stated in detail previously [16].

### 4.3. pH Assessment

We measured the pH by a single-use pH catheter (Medical Measuring Systems (MMS) Greenfield, MMS pH with one antimony channel), a flexible wire protected by a sterile plastic cover. The system obtained recordings with a frequency of 1 Hz, stable values within few seconds, and an accuracy of 0.1 pH units. Prior to each examination, the sterile catheter was calibrated (buffer pH 4.00 and buffer pH 7.00) at room temperature. With a sterile speculum, the vaginal wall and the external os of the cervix were visualized, the pH catheter was applied with a sponge holder at the hymen rim, moved along the vaginal wall to the posterior fornix, through the cervical canal, and in the nonpregnant participants further up to the upper uterine cavity (Figure 1). Resistance occurs between the upper cervical canal and the uterine cavity, marking the measuring point in this region. The pH catheter was equipped with centimeter marking ensuring anatomical accuracy in every measurement (Figure 1). Measuring points were picked as introitus (IN) (1–2 cm from the external os), mid-vagina (MV) (approximately 5 cm from external os), posterior fornix (FP) (behind the cervical mouth), distal cervical canal (DC) (just inside the cervix), proximal cervical canal (PC) (measured prior to resistance point of the uterine cavity), lower uterine cavity (LU) (after resistance point), and upper uterine cavity (UU) (ultrasonically verified) (Figure 1).

### 4.4. Statistical Methods

Box-and-whisker plots of the pH values measured along the genital tract in women at different stages of pregnancy (A) and in nonpregnant (B) and early-pregnant (C) women divided according to the identified vaginal microbiome. The horizontal line is plotted at the median, the box extends from the 25th to the 75th percentile, and the whiskers identify the maximum and minimum values. Data were compared by repeated measure two-way ANOVA followed by Sidak’s post hoc tests: * *p* < 0.05, ** *p* < 0.01, *** *p* < 0.001 vs. NVM or as indicated.

## 5. Conclusions

In nonpregnant and pregnant women with normal vaginal microbiota, we observed a pronounced pH gradient in the genital tract. Women with abnormal vaginal microbiota had significantly higher pH in the lower vagina but surprisingly not in the upper vagina, the cervical canal, or in the uterus.

## Figures and Tables

**Figure 1 pathogens-10-00090-f001:**
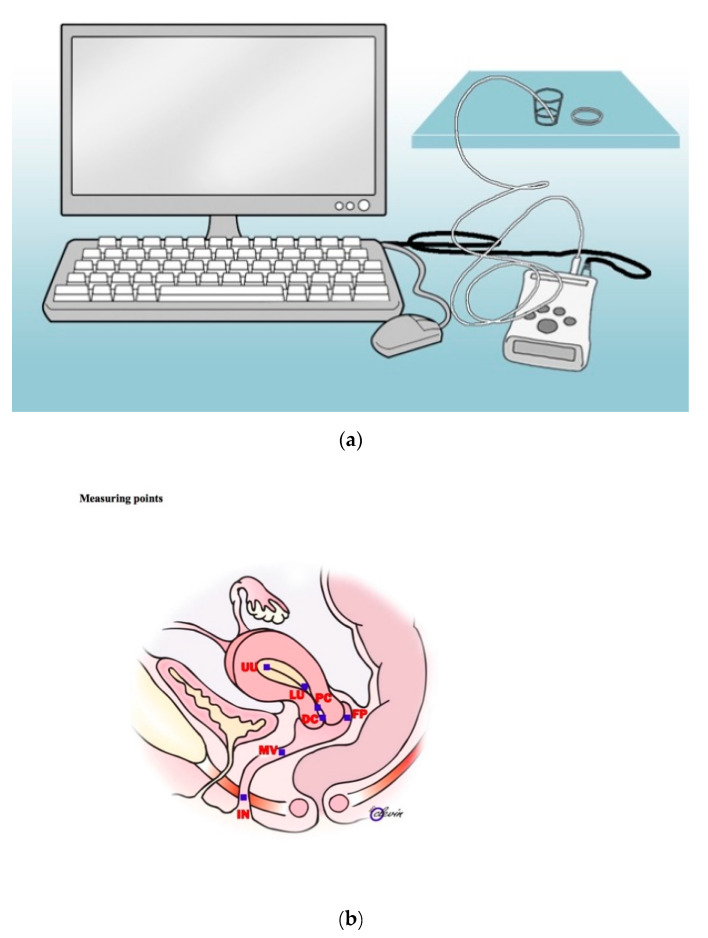
Technical illustrations of the calibration procedure (**a**) and measurement points (**b**) used for evaluating pH along the female genital tract. In (**b**) pH measuring points are market with a blue square, IN = introitus, MV = mid-vagina, FP = fornix posterior, DC = distal cervical canal, PC = proximal cervical canal, LU = lower uterine cavity, and UU = upper uterine cavity.

**Figure 2 pathogens-10-00090-f002:**
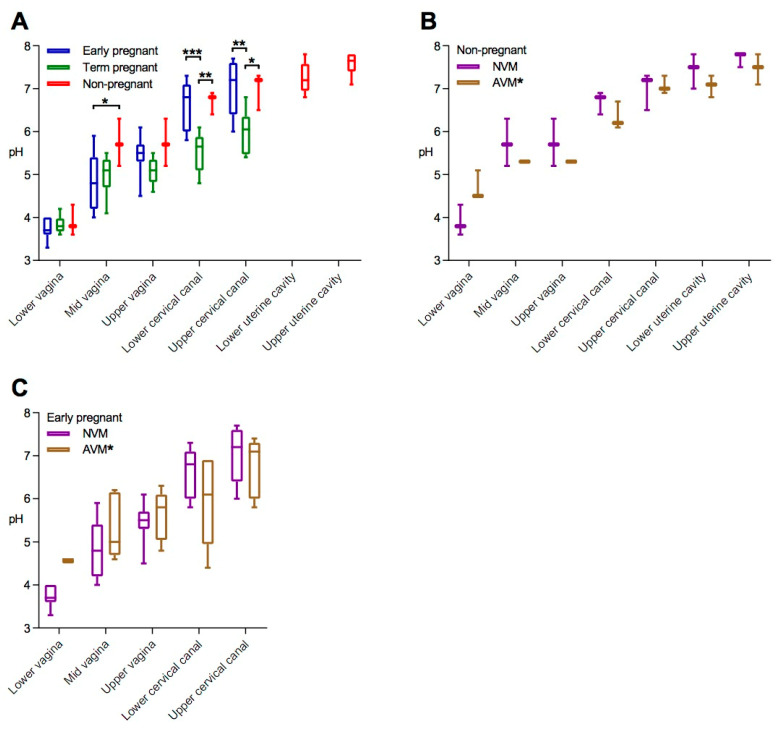
Box-and-whisker plots of the pH values measured along the genital tract in women at different stages of pregnancy (**A**) and in nonpregnant (**B**) and early-pregnant (**C**) women divided according to the identified vaginal microbiome. The horizontal line is plotted at the median; the box extends from the 25th to the 75th percentile; and the whiskers identify the maximum and minimum values. Data were compared by repeated measure two-way ANOVA followed by Sidak’s post hoc tests: * *p* < 0.05, ** *p* < 0.01, *** *p* < 0.001 vs. normal vaginal microbiota (NVM) or as indicated.

## Data Availability

No new data were created or analyzed in this study. Data sharing is not applicable to this article.

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
