# Peer review of "Vaginal, Cervical and Uterine pH in Women with Normal and Abnormal Vaginal Microbiota"

_pathogens, 2021, doi:10.3390/pathogens10020090_

Round 1

Reviewer 1 Report

The study “Vaginal, cervical and uterine pH in women with normal and abnormal vaginal microbiota” by Lykke et al., has some strengths and limitations. This study is timely. However, the manuscript needs major clarification. In the comments to the authors, I have some suggestions for its improvement. In my opinion, this manuscript needs to be improved and clarified to be published in Pathogens.

Weaknesses of the study:

The manuscript has a major weakness regarding the qPCR experiments. The authors described that performed such experiments, but specific information regarding the procedure and the results (the bacterial load/cycle of Gardnerella/A. vaginae) is not described. Furthermore, the data regarding the Nugent score is also not shown. As this is a small manuscript, the authors need to complete it and show all the results.

Line 4. The authors should introduce that BV is the most frequent vaginal infection in women of reproductive age.

Line 57/71. Lactobacillus should be formatted in italic.

Line 73 and the rest of the manuscript - the name of the different bacterial species should be formatted in italic.

Line 72. The authors must need to change “Gardnerella vaginalis” by “Gardnerella”. G. vaginalis was the only recognized species in its genus for four decades, but recently an emended description of G. vaginalis and descriptions of three new species – Gardnerella leopoldii, Gardnerella piotii, and Gardnerella swidsinskii – have been proposed. See the following manuscripts. Trends Microbiol. 2020 Mar;28(3):202-211. doi: 10.1016/j.tim.2019.10.002. https://doi.org/10.1099/ijsem.0.003200

Line 108. “n  referring  to  number  of  number  of” sounds awful.  

Line 110. Please clarify which test was used to confirm that the data follow a normal distribution in order to be correctly used two-way ANOVA test.

Line 121. “CO2” Please format the chemical formula.

Line 184. “5oC” Please uniformize the style of the temperatures.

Section 183-190. The authors must need to improve this section. Describe the protocol, primers used, and specific controls. Of note that we cannot differentiate Gardnerella vaginalis from the other species by 16S rRNA. As such, the authors should take into consideration this information.

Author Response

Dear reviewer.

Thank you for your comments and suggestions. We have changed the minor errors within the manuscript (see attached), and added the following:

Line 4. The authors should introduce that BV is the most frequent vaginal infection in women of reproductive age.

Answer: We have added a few lines at the beginning of the introduction concerning this relevant observation. 

Section 183-190. The authors must need to improve this section. Describe the protocol, primers used, and specific controls. Of note that we cannot differentiate Gardnerella vaginalis from the other species by 16S rRNA. As such, the authors should take into consideration this information.

Answer: All of our qPCR results have been combined with Nugent classification and pH values in appendix 1. 

Reviewer 2 Report

This manuscript by Lykke et al., analyzes variations in pH and microbial composition in different compartments of the genital tracts of non-pregnant and pregnant women. While some of the findings are interesting, there are some significant issues as follows:

  1. While the authors describe having performed real time PCR to quantify microbes in the genital tracts of the sampled individuals, those results are not included in the manuscript nor are they described any detail.
  2. They report variations in pH among different groups. Such changes in pH from normal are often associated with changes in genital tract microbial composition. It would, therefore, be important to include microbial composition data and relate that to the pH variations.

Minor comments:

  1. Line 61: change “stabile” to “stable”
  2. Why is Figure 1 presented with two legends?
  3. Line 107: delete one phrase “number of”
  4. Line 141: delete “in”
  5. Line 151: delete “can”

Author Response

Dear reviewer.

Thank you for your comments and suggestions. Below we have answered your remarks one by one:

  1. While the authors describe having performed real time PCR to quantify microbes in the genital tracts of the sampled individuals, those results are not included in the manuscript nor are they described any detail

  2. They report variations in pH among different groups. Such changes in pH from normal are often associated with changes in genital tract microbial composition. It would, therefore, be important to include microbial composition data and relate that to the pH variations.

Answer: A table with qPCR results, Nugent classification and pH has been drafted and is now found as "appendix 1"

Answer to your minor comment are to be found within the newly uploaded manuscript.

Reviewer 3 Report

In this study the Authors measured pH in different compartments of the female genital tract in both non-pregnant and pregnant women, stratifying into a normal and abnormal vaginal microbiota. The manuscript has some merit and some novelty. However, I have many perplexities.

My concerns regard especially the design of the study and statistical methods, but there are also other points that need to be clarified.

Pag. 2. line 57

The Authors evaluated six study groups: non pregnant, early and term pregnant, with normal and abnormal vaginal microbiota. However, they enrolled only 26 women, including, consequently, very few subjects in each group. Did the Authors calculate the sample study? It is well known that determining the right sample size is crucial for strong experimental design.

Pag. 2, line 65

Only 3 non-pregnant women with NVM were evaluated. This sample is too small to draw reliable conclusions.

Pag. 3

Figure 2 is complex and difficult to understand. There is no clear connection between the data showed in the figure and those reported in the Results section. I suggest to add a table, reporting all data with p-values, and to discuss the results point by point.

Moreover, I think that the Authors should provide the raw data in the Appendix section.

Pag. 4, line 89

Statistics methods should be reported in the Methods section.

Pag. 4, line 94

The Authors stated: “This study demonstrated a rather dramatic pH-gradient in the female genital tract.”  However, are all differences statistically significant?

“Remarkable” “Dramatic increase” should be replaced with significant or extremely significant where appropriate.

Pag. 5, line 143

The authors state: “Thirdly, one can hypothesize that the moving direction of the spermatozoa in their journey from the upper vagina to the uterus towards the oocyte is controlled by the pH-gradient.”

This is a very interesting hypothesis. However, if true, significant changes of pH gradient before and after the ovulation should be detected, like the uterine contractility gradient. A larger study involving many women in different phase of menstrual cycle is necessary to suggest this hypothesis.

Pag. 5, line 165

To obtain more reliable findings, all measuring points of proximal cervical canal and uterine cavity should be ultrasonically verified.

Did the catheter insertion in the uterine cavity cause bleeding? This is a very frequent event during uterine cavity device insertion and it could have influenced the measurements. A large sample study could exclude this bias.

Pag. 5, line 177

The Authors should stress that further studies involving a large number of subjects are necessary to draw more firm conclusions.

Pag. 5, line 181

The authors state: “Thus, the pH does not seem to be causally involved in the pathophysiological mechanisms of infertility and preterm birth.” The objective of this study was not to determine the role of pH in infertility and preterm birth. Moreover, the results of the study do not support these conclusions.

Author Response

Reply to reviewer:

Thank you for your comments and suggestions. Below you will find our answers and corrections.

The Authors evaluated six study groups: non pregnant, early and term pregnant, with normal and abnormal vaginal microbiota. However, they enrolled only 26 women, including, consequently, very few subjects in each group. Did the Authors calculate the sample study? It is well known that determining the right sample size is crucial for strong experimental design.

Answer: We did not conduct a formal sample size calculation. To clarify further, we have conducted a table with qPCR results, Nugent classification and pH measurements, attached as appendix 1.

Pag. 2, line 65

Only 3 non-pregnant women with NVM were evaluated. This sample is too small to draw reliable conclusions.

Answer: We agree with the reviewer on this matter and has commented to this challenge in the “discussion”.

Pag. 3

Figure 2 is complex and difficult to understand. There is no clear connection between the data showed in the figure and those reported in the Results section. I suggest to add a table, reporting all data with p-values, and to discuss the results point by point.

Moreover, I think that the Authors should provide the raw data in the Appendix section.

Answer: Any significant result is mentioned in the “Results” section. Please see additional details in appendix 1 attached.

Pag. 4, line 89

Statistics methods should be reported in the Methods section.

Answer: This has been corrected in the resubmitted paper at the end of the methodical session.

Pag. 4, line 94

The Authors stated: “This study demonstrated a rather dramatic pH-gradient in the female genital tract.”  However, are all differences statistically significant?

“Remarkable” “Dramatic increase” should be replaced with significant or extremely significant where appropriate.

Answer: Please see “Results” section 2 for further details concerning the significant findings of this study. When using the word “significant” we think of it as referring to statistical calculations. Therefore, we have chosen to change the words “remarkable” and “dramatic” when used to “striking” and “pronounced” respectively.

Pag. 5, line 143

The authors state: “Thirdly, one can hypothesize that the moving direction of the spermatozoa in their journey from the upper vagina to the uterus towards the oocyte is controlled by the pH-gradient.”

This is a very interesting hypothesis. However, if true, significant changes of pH gradient before and after the ovulation should be detected, like the uterine contractility gradient. A larger study involving many women in different phase of menstrual cycle is necessary to suggest this hypothesis.

Answer: Thank you. We believe we can add such a hypothesis in the discussion section, but we also agree with the reviewer concerning the fact that further studies are needed to clarify this thought further. We have added a comment in the discussion.

Pag. 5, line 165

To obtain more reliable findings, all measuring points of proximal cervical canal and uterine cavity should be ultrasonically verified.

Answer: Yes, we absolutely agree. We performed ultrasonic measurements during the first two tests, but it wasn’t always possible to have access to ultrasound during each procedure, since this wasn’t standard during the surgical procedure a note about this has been added to the methodical section.

Did the catheter insertion in the uterine cavity cause bleeding? This is a very frequent event during uterine cavity device insertion and it could have influenced the measurements. A large sample study could exclude this bias.

Answer: We did experience contact bleeding during pH-measurements, but as stated in material and methods, participants who experience any kind of bleeding (that being prior to or during the investigation) were excluded from the final data.

Pag. 5, line 177

The Authors should stress that further studies involving a large number of subjects are necessary to draw more firm conclusions.

Answer: This has been added to the discussion.

Pag. 5, line 181

The authors state: “Thus, the pH does not seem to be causally involved in the pathophysiological mechanisms of infertility and preterm birth.” The objective of this study was not to determine the role of pH in infertility and preterm birth. Moreover, the results of the study do not support these conclusions.

Answer: As stated in the introduction BV is thought to affect the female genital canal in many ways, there among pregnancy. Therefore, we thought of the above as a supplemental thought to our findings: that pH does not seem to be involved in the pathophysiological mechanisms of infertility and preterm birth. But we also recognise the scope of this paper and have therefore chosen to remove the sentence from the discussion.

Round 2

Reviewer 1 Report

The authors carefully addressed the questions. However, some points need clarification.  

Major concerns:

Line 203. "These four bacterial species were found to be the optimal bacteria for predicting BV". The authors should clarify why they do not include P. bivia since the combination of Gardnerella spp., A. vaginae and P. bivia is considered a strong marker of BV.

Line 204. "All samples were run in the same lab and with the same methods as described in the paper by Datchu et al". The authors should briefly describe the methods used by Datchu et al. as well as explain how they obtained the results from the qPCR in the supplementary material file. Additionally, in line 206, the authors claim that qPCR was performed for the four common Lactobacillus spp. (Lactobacillus crispatus, Lactobacillus gasseri, Lactobacillus iners, and Lactobacillus jensenii). As such, the authors should provide this data.

Author Response

Dear reviewer.

Thank you for your comments. Please see the answers to your notes below.

Line 203. "These four bacterial species were found to be the optimal bacteria for predicting BV". The authors should clarify why they do not include P. bivia since the combination of Gardnerella spp., A. vaginae and P. bivia is considered a strong marker of BV.

Answer: The Prevotella probe used is a genus-wide probe covering multiple Prevotella species, including P. bivia (REF. Datcu et al.). This has been added to the manuscript (line 219).

Line 204. "All samples were run in the same lab and with the same methods as described in the paper by Datchu et al". The authors should briefly describe the methods used by Datchu et al. as well as explain how they obtained the results from the qPCR in the supplementary material file. Additionally, in line 206, the authors claim that qPCR was performed for the four common Lactobacillus spp. (Lactobacillus crispatus, Lactobacillus gasseri, Lactobacillus iners, and Lactobacillus jensenii). As such, the authors should provide this data.

Answer: Thank you for your comment. We hereby provide a brief description which has also been amended to the manuscript (line 257).

We agree, it is not correct. We do not provide data for the four common Lactobacillus spp. This has been revised.

Reviewer 2 Report

The newly added Appendix 1 is not stated in the manuscript text.

Author Response

Dear reviewer.

The newly added Appendix 1 is not stated in the manuscript text.

Answer: Thank you for correcting this error. This has now been added to the text, line 97.

Reviewer 3 Report

Thank you for accepting the comments and suggestions. The paper was improved but there are still few points to be revised:

Line 40

“dramatic” should be replaced with “pronounced”.

Line 90-91

The range of the values should be added.

Line 169

The sentence “This would cause the evidence of pH changes before and after ovulation and to support it, a uterine contractility gradient.” is not clear.

Line 234

The sentence “Thus, the pH does not seem to be causally involved in the pathophysiological mechanisms of infertility and preterm birth.” should be removed.

Author Response

Dear reviewer.

Thank you once again for your comments. The following has been corrected/added:

Line 40

“dramatic” should be replaced with “pronounced”. This has been corrected.

Line 90-91

The range of the values should be added.

Answer: This has now been added

Line 169

The sentence “This would cause the evidence of pH changes before and after ovulation and to support it, a uterine contractility gradient.” is not clear.

Answer: This sentence has been added in correspondence with another reviewer. However, we have chosen to correct it slightly for clarification. 

Line 234

The sentence “Thus, the pH does not seem to be causally involved in the pathophysiological mechanisms of infertility and preterm birth.” should be removed.

Answer: thank you for this comment. This has now been revised.

Round 3

Reviewer 1 Report

The authors carefully addressed each question, being the paper now suitable for publication.